# Role of Ceramides and Lysosomes in Extracellular Vesicle Biogenesis, Cargo Sorting and Release

**DOI:** 10.3390/ijms232315317

**Published:** 2022-12-05

**Authors:** Rostyslav Horbay, Ali Hamraghani, Leonardo Ermini, Sophie Holcik, Shawn T. Beug, Behzad Yeganeh

**Affiliations:** 1Apoptosis Research Centre, Children’s Hospital of Eastern Ontario Research Institute, Ottawa, ON K1H 8L1, Canada; 2Centre for Infection, Immunity and Inflammation (CI3), University of Ottawa, Ottawa, ON K1H 8L1, Canada; 3Department of Cellular and Molecular Medicine, University of Ottawa, Ottawa, ON K1H 8M5, Canada; 4Department of Life Sciences, University of Siena, Via Aldo Moro 2, 53100 Siena, Italy; 5Department of Biochemistry Microbiology and Immunology, Faculty of Medicine, University of Ottawa, Ottawa, ON K1H 8L1, Canada

**Keywords:** exosome, multivesicular body, intraluminal vesicles, mitochondria-derived vesicle, ESCRT-dependent pathway, ESCRT-independent pathway

## Abstract

Cells have the ability to communicate with their immediate and distant neighbors through the release of extracellular vesicles (EVs). EVs facilitate intercellular signaling through the packaging of specific cargo in all type of cells, and perturbations of EV biogenesis, sorting, release and uptake is the basis of a number of disorders. In this review, we summarize recent advances of the complex roles of the sphingolipid ceramide and lysosomes in the journey of EV biogenesis to uptake.

## 1. Introduction

Extracellular vesicles (EVs) are secretory components that are well known as biologic communicatory systems that contain multiple signaling molecules, including RNAs, proteins, and lipids. Based on the size and origin of biogenesis, EVs are classified into three groups: exosomes, microvesicles (MVs) and apoptotic bodies [1]. During multivesicular body (MVB) formation, depending on nature of the cargo that is packaged into the early endosome, intraluminal vesicles (ILVs) can end up for targeted cargo degradation or be released as exosomes from the cell [2]. Both mechanisms are regulated by endosomal sorting complex required for transport (ESCRT)-dependent and -independent mechanisms, and these vesicles are in a crosstalk with lysosomes, autophagosomes, Golgi apparatus, mitochondria and other organelles [3,4]. Lysosomes are the main recipients of ILVs trapped inside MVBs. These organelles can degrade the inner content of the multivesiclar body or release exosomes during lysosomal exocytosis [5]. Ceramides have a key role in EV biogenesis. Ceramides are a diverse family of structural and bioactive lipid molecules that are essential for cell function [6]. As ceramides are the building blocks of bilayers membrane in all vesicles, many researchers have investigated the role of ceramides in EVs biogenesis. In this review we will focus on the role of ceramides in EV biogenesis, release and their effects in recipient cells.

## 2. Ceramides

### 2.1. Ceramide Structure

Ceramides are the principal sphingolipid metabolites composed of a Fatty Acid (FA) linked via an amide bond to a long-chain aliphatic amino alcohol or sphingoid base (sphingosine, phytosphingosine, dihydrosphingosine, or 6-hydroxysphingosine). Ceramides are a diverse family of structures that vary on FA chain length (medium chain, C12 to C14; long chain, C16 to C18; very-long chain, C20 to C24; ultra-long chain, ≥C26) and saturation, as well as by the length of the sphingoid base and the modification such as the presence of hydroxyl group on sphingoid backbone [7]. To date, more than 1500 structurally different ceramides and additional ceramide subclasses have been identified and many more are currently discovered due to continuous advances in mass spectrometry approaches and protocols [8,9].

### 2.2. Biogenesis of Ceramides: Role of Endoplasmic Reticulum, Golgi and Lysosome

Sphingolipid metabolism is extremely complex and is regulated by the activity of more than 40 enzymes [10]. Ceramide is the core of sphingolipid metabolism and can be produced or utilized through several pathways. There are three main synthesis pathways of ceramide: the de novo pathway, the sphingomyelinase (SMase) pathway (or catabolic pathway), and the salvage pathway (Figure 1).

The de novo pathway originates in the endoplasmic reticulum by the condensation of serine and palmitoyl CoA, catalyzed by serine palmitoyltransferase (SPT), to produce 3-ketodihydrosphingosine that is then converted by the 3-ketodihydrosphingosine reductase to dihydrosphingosine (sphinganine). Sphinganine is then acylated to dihydroceramide by ceramide synthase, which is later desaturated by dihydroceramide desaturase to form ceramides [11]. Ceramides can be delivered from ER to the Golgi apparatus either by a vesicular route (vesicular transport) or nonvesicular transport by specific translocation via the action of ceramide transfer protein (CERT) [12]. Ceramide transported to the Golgi complex is further modified by the addition of several head groups to form different classes of complex sphingolipids such as sphingomyelin (ceramide from vesicular transport) and glycosphingolipids (ceramide from nonvesicular transport or CERT) [10]. In the Golgi compartment, ceramides that serve as the substrate for sphingomyelins flip across membrane leaflets whereas the synthesis of glucosylceramide from ceramide happens on the cytosolic side of the Golgi and then glucosylceramides flip for the synthesis of higher glycosphingolipids [13]. However, the machinery involved in this flipping is not yet well defined. Sphingomyelin (major product of ceramide conversion) is obtained by the activity of sphingomyelin synthase enzymes SMS1 and SMS2 [14]. SMS1 is a Golgi enzyme and SMS2 is present in the plasma membrane. The lipids are then transferred to the plasma membrane (the major cellular reservoir) via the vesicular transport pathway or lipid-transfer proteins.

The SMase pathway is a sphingolipid catabolic pathway that produces phosphocholine and ceramide at the plasma membrane of cells or in lysosomes by the activity of SMase [15]. Constitutive catabolism occurs in the late endosomes and lysosomes where ceramides can be additionally metabolized by acid ceramidases to generate sphingosine (Sph) and free FA by the enzymatic activity of ceramidases [16]. FAs and sphingosine can then leave the lysosome and Sph can be phosphorylated by Sph kinases (SphK) to form sphingosine-1 phosphate (S1P) [17]. Ceramide in the lysosomal compartment can also be produced from the breakdown of globoside by hexosaminidase from sulfatide that loses a sulfate group by arylsulfatase A and from ganglisoides GM1 that loses one unit of galactose. Ceramides, FAs and sugar (obtained by the catabolism of glycosphingolipids) are transferred from the lysosomes through specific membrane proteins and reach once again the Golgi apparatus where they are used in the biosynthetic pathway [18]. Several reports have revealed the role of lysosomes for sphingolipid metabolism and its importance for cell physiology [17]. In particular, many researchers have focused their studies on the metabolism of ceramide in lysosome and sphingolipid storage disorders such as Sandhoff disease, Fabry disease, Gaucher disease, Krabbe disease, Niemann Pick disease, Farber disease and preeclampsia [19,20].

Ceramides can also be generated by “salvage pathway or recycling pathway” in the late endosomes and lysosomes through recycling of sphingosine. In the salvage pathway, the long-chain sphingoid base sphingosine produced from the metabolism of complex sphingolipids is converted to ceramide through the action of a number of enzymes including SMases, cerebrosidases, ceramidases, and ceramide synthases [13].

### 2.3. Physiopathological Function of Ceramides

Sphingolipids are versatile molecules that have vital roles as structural components of cell membranes and as signal transduction molecules involved in regulation of many of the most critical cellular events, including growth, differentiation, programmed cell death and tumorigenesis. Ceramides are fundamental precursors of complex sphingolipids and components of cell membranes. Ceramides, in particular, long chain and very-long chain ones are the major lipid class present in the human stratum corneum barrier, the remotest layer of the epidermis that safeguards underlying tissue from mechanical stress, dryness, illness, and chemicals [21]. Ceramides are also the main component of a particular group of lipid membrane microdomains termed “ceramide-enriched membrane domains” that are involved in the induction and magnification of receptor and stress-mediated signaling in the majority of cell types [22]. More fundamentally, ceramides are also important bioactive lipids involved in cellular regulatory circuits. Indeed, the heterogeneity of ceramides, as well as the flexibility of sphingolipid metabolism, suggests a variety of possible mechanisms to regulate cell fate and functions. Several recent pieces of evidence have indicated that ceramides are critical mediators of various cell death pathways, including apoptosis, autophagy, necrosis, and necroptosis [10,20,23,24].

Altered levels of ceramides trigger signaling pathways leading to cell death in various cell types. For instance, elevation of ceramide levels in the lung result in endothelial cell death, which is linked to the development of emphysema-like disease in murine models [25]. Rotstein et al. showed that ceramides would provoke apoptosis and inflammation in endothelial and retinal pigmented epithelium cells, which bring to several retinopathies [26]. Similarly, treatment of fetal rat lung epithelial cells with C16 ceramide induces autophagy and apoptosis in a temporal pattern [27]. Recent research has also indicated that ceramide can selectively regulate cell fate differently in the placenta trophoblast layers. High levels of ceramide bring excessive autophagy that leads to Type II cell-death in syncytiotrophoblast layer [20,23]. Instead, in the cytotrophoblast layer the sphingolipid accumulation can activate the homeostatic process of mitophagy or caspase-independent type III programmed cell death [28].

Altered ceramide metabolism has been documented in several types of cancers [29] and several studies suggest that there are diverse and controversial functions of endogenously generated ceramides in malignant cells. For example, C16 ceramide has been implicated in cancer cell proliferation, whereas C18 ceramide mediates cell death [30]. Due to its importance in tumorigenesis, some of the metabolic enzymes of ceramide synthesis are the targets of current chemotherapy drugs [31].

Ceramides can regulate a variety of pathways that promote inflammation. Ceramide inhibits the Akt pathway by activation of PP2A or PKCζ, thereby blocking insulin signaling which leads to insulin resistance and decreased cell survival [32,33]. Moreover, ceramides can influence innate immune responses regulating autophagy, promote the processing and secretion of the pro-inflammatory cytokines IL-1β and IL-18, and induce apoptosis [34]. Furthermore, numerous age-related, neurological, and neuroinflammatory diseases are shown to be regulated by intracellular levels of ceramide [35].

Several pieces of evidence have indicated that ceramide levels are important for lung development to maintain lung cell homeostasis and correct host response to airway microbial infections [36]. Furthermore, the lipids play a pivotal role in pulmonary fibrosis, chronic obstructive pulmonary disease (COPD), and asthma [37]. Ceramide accumulation in blood vessels and heart is present in subjects with cardiovascular disease (such as hypertension and heart failure) [38] and plasma and placentae from a pregnancy complicated by preeclampsia [20].

## 3. Extracellular Vesicles (EVs)

### 3.1. EV Biogenesis and Cargo Loading

EVs represent a heterogenous population of vesicles that are classified into three groups typically based on their size and biogenesis: exosomes (30–200 nm), MVs (100–1000 nm) and apoptotic bodies (>1000 nm) [1]. Here, we use EV as generic term for all secreted vesicles from cells, since most research described in the literature has employed heterogeneous populations of EVs and often failed to fully characterize the isolated EVs [1,39]. Although several molecular families and organelles have been shown to have important roles in the EV biogenesis (e.g., Rab, ESCRT, ceramide, syntenin, syndecans, tetraspanins, etc.), trafficking (e.g., Rab, actin, etc.), and fusion of MVBs with the plasma membrane (e.g., SNAREs), the focus of this review is on sphingolipid ceramide as one of the lipids critical for EV formation as well as lysosomes as the main organelle for the biogenesis, cargo loading and release of EVs.

Even though the most distinguishable hallmark of EV biogenesis is the involvement of the ESCRT pathway, EVs can be generated through an ESCRT-independent pathway. Packaging of endocytosed parts of plasma membrane, receptors, encapsulated ubiquitinated proteins, and accumulation of those in the form of ILVs ensures regulation of cell signaling and transition of the early endosome into the late endosome, also known as MVB (Figure 2). This mechanism is regulated by the classical ESCRT-dependent mechanism, which is also involved in EV release. Damaged protein cargo coming from the plasma membrane and Golgi apparatus is packaged into the MVB inside ILVs. The typical fate of the MVB is its fusion with the lysosome [40,41]. The ESCRT pathway is involved in many processes, including cellular abscission, viral budding, and even daughter cell separation during cell division in eukaryotic cells [40,42,43]. Of these processes, one that stands out is the EV release through the ESCRT pathway, and is predominantly associated with packaging and recycling ubiquitylated proteins [41,44]. However, SMase, tetraspanins and other supporting proteins are involved in EV biogenesis that are not restricted to ESCRT only EV biogenesis [6,45]. Furthermore, ceramide is considered the main building block of EVs and can be involved in both ESCRT-dependent and -independent pathways. The ceramides involved in EV biogenesis pathways are less strictly ESCRT linked and mostly associated with direct membrane budding [3,6,46]. Both pathways can be observed in metazoa (which we will use as an example), yeast and archaea, but the nature of the cargo is not the same [42,45,47].

#### 3.1.1. ESCRT-Independent Mechanism

MVB biogenesis and EV formation can occur without the presence of ESCRTs in a ubiquitin-independent manner. Despite simultaneously silencing all four ESCRTs, EV generation can still occur [3]. One of the many key regulators of ESCRT-independent EV pathway is Ras-associated binding 31 (Rab31) (Figure 3). Once Rab31 is phosphorylated by epidermal growth factor receptor (EGFR), it will let flotillin proteins traffick EGFR inside the MVB for packaging into ILVs in an ESCRT-independent manner [46]. Flotillins will engage in lipid rafts to drive the inclusion of EGFR into MVBs in order to form ILVs in an ESCRT-independent manner. Rab31 recruits TBC1D2B (a GTPase activating protein) in order to inactivate Rab7 [46], a small GTPase key regulator of transport to late endocytic compartments such as late endosomes and lysosomes [48]. Once Rab7 is inactive, lysosomal degradation of MVBs is interrupted and enables the release of ILVs from the cell [46]. Rab31 is thus essential increased EV release, as it can recruit TBC1D2 to inactivate Rab7. This will prevent MVB-lysosome fusion and will result in ILV release [46]. Another study has shown that sphingholipid ceramide can be enriched in EVs that are generated in an ESCRT-independent manner [6]. However, the release of such EVs in an ESCRT-independent manner was significantly reduced upon inhibiting neutral sphingomyelinase (nSMase), an enzyme that generates ceramides from sphingomyelin [6]. Several proteins including tetraspanins Tspan8, CD9, Rab proteins, flotillin-1 and other classical EV biomarkers are shared between ESCRT-dependent and -independent pathways [3,46,49]. Some of the most prominent hallmarks that distinguish between the two pathways is the packaging of RNA and the use of ceramides as building blocks for EV generation [47]. A greater proportion of lipids in EV formation can represent another hallmark of the ESCRT-independent EV biogenesis pathway [6]. The cargo was secreted into a distinct subdomain of the endosomal membrane that required ceramides in ESCRT-independent manner. The application of a nSMase2 inhibitor such as GW4868 reduces the number of secreted EVs due to the lack of core building blocks for the vesicles, including the proteolipid protein [6,50]. While nSMase2 converts sphingomyelin into ceramide, the main building block of EVs [6], ceramide blockage is responsible only for a fraction of EV release [3,6,51]. Furthermore, EV release can be both ceramide and nSMase2 independent [52,53]. Interestingly, the lipidome of the EVs can distinguish drastically from larger vesicles, such as MVs. MVs will have a lipidome more of a plasma membrane, whereas EVs are more rigid, as they contain lipid rafts, phospholipases A2, C, D. In addition they are enriched with common EV biomarkers heat shock proteins-70 and -90 [54,55].

LC3 is a central protein in the autophagy pathway that can selectively recruit cargo to the autophagosome via interacting with cargo receptors. This protein can selectively incorporate RNA binding proteins (RBP) and non-coding RNA into extracellular vesicles in an ESCRT-independent manner. Such LC3-dependent extracellular vesicle loading starts with LC3 positioning on the outside of the MVB. Once on the MVB, three proteins—SMPD3, NSMAF and RBP—will surround LC3 in order for LC3 to undergo intraluminal budding. This mechanism will drive ceramide production at the MVB membrane to facilitate EV biogenesis [56].

#### 3.1.2. ESCRT-Dependent Mechanism

The ESCRT-dependent EV biogenesis is an evolutionary conserved pathway, mainly regulated by Ubiquitin (Ub) which mediates degradation of a wide variety of membrane proteins in lysosomes by sorting them into the ILVs of MVBs [4]. There are 4 ESCRT complexes, ESCRT-0, -I, -II, and -III, that are accompanied by accessory proteins such as the Vps4-Vta1 ATPase in complex with Alix. Alix is involved in ILV generation and packaging into the MVB, cargo packaging into ILVs, and ILV/exosome release from the cell (Figure 4). Each stage decides the fate of the ILVs, whether they will be released as EVs or undergo lysosomal protein degradation [51]. ESCRT-0 is absent in some protists and plants, and is not necessary for initial MVB formation. The main role of this pathway is to safely deliver utilize the ubiquitinated protein cargo (Ub-cargo) for degradation [44,57]. The early endosomal protein Hepatocyte growth factor-regulated tyrosine kinase substrate (Hrs) and the signal transducing adaptor molecules (STAMs) each contain a Ub-interacting motif. Hrs and STAMs form a complex that leads to transport of early endosomes. Ub-cargo will cluster along the endosomal membrane in order to be part of the ESCRT-0, a Vps27-Hse1 heterodimer complex in yeast, or its mammalian analogue Hrs/STAM1 [58]. Once Ub-cargo is localized along the early endosome, the heterodimer Hrs-STAM1/2 will complex along Eps15 and the Ub-cargo to form the ESCRT-0 complex [57]. Subsequently, after the ESCRT-0 complex is generated a vesicular envelope containing clathrin will be formed [59]. Such localization of clathrin, STAM2, Hrs and Eps15 is controlled by MVB forming protein Vps4 [59]. Vps4 presence is important for tetraspanin vesicle formation [60].

The whole transition of unwanted cargo from its internalization by the cell until exosome release or lysosomal degradation by the ESCRT-machinery, several transition steps are involved during each ESCRT-complex formation, The transition of ESCRT-0 to ESCRT-I is tightly regulated by Flotillin-1, as this protein regulates TSG-101 activity, cargo recognition and sorting during the transition [61,62]. TSG-101, a regulator of vesicular trafficking, binds to Ub-cargo proteins, as well as is a key player in sorting this cargo into the MVB [63]. Hrs/Vps27 will bind directly to the amino terminus of the TSG101/Vps23 subunit of ESCRT-I to further promote endosomal formation. The ESCRT-I subunits are essential for MVB generation and its stabilization [61,64]. A proline rich motif of Vps23 directs ESCRT-I towards the midbody during membrane scission once it meets ESCRT-III. TSG-101 will guide the ESCRT-I complex and favor its interaction with ESCRT-0, Ub-cargo [64], and the GRAM-like Ub-binding in Eap45 (GLUE) protein from ESCRT-II will bridge via NZF with Ub-associated protein 1 (UBAP1), a component of ESCRT-I complex [65], while GLUE and UBAP1 both bind to different Ub-cargo. Without each of the previous ESCRT steps, the budding of membrane into lumen will not occur and subsequently would block ESCRT-III formation and ILV scission [61,64]. Due to the presence of Vps23, ESCRT-III regulates all previous ESCRT complexes. As a transient complex, ESCRT-III is the most important ESCRT-complex, as it is the final step where ILVs are released as exosomes or the cargo is shuttled to the lysosome for degradation [44,66]. Upon anchoring of Vps20 to the membrane surface, Alix directly binds to CHMP4 [67]. Subsequently, after spiralization and convexity of the polymerized CHMP4 is formed, Vps24 and Vps2 subsequently pinches off the vesicle [51,68,69]. Alix, along with Syntenin and Syndecans, two other key regulators of endocytosis and release of exosomes, thus regulate EV biogenesis [63]. This is a stage where the Ub-cargo can deubiquitinated by an Ub hydrolase Doa4, which is also recruited by Alix, to bind to Snf7 on ESCRT-III, and leads to restoration of the MVB to potentially enable EV release instead of shuttling cargo to the lysosome [70,71]. Later the ESCRT-III complex will be dissembled by the Vps4/Vta1 complex, recycled and ready for the next cycle [72]. It worth mentioning that tetraspanins, such as the classical exosomal biomarkers CD63, CD81 and CD82, enable vesicle budding by creating membrane curvatures and favoring cargo sorting into vesicles [42,59,73]. While EV biogenesis is often described as ESCRT-dependent or -independent, these two pathways might not be wholly independent [74]. Exosomal release could be a synergistic co-release by several pathways, which could even ensure a backup mechanism for the unique and more prominent ones.

### 3.2. Mitophagy and Mitochondria-Lysosome Cargo Shuttling Route

As the “cell’s engine”, mitochondria can undergo fusion and fission in order to maintain quality control and prevent unnecessary inflammation [75]. Mitochondrial fusion is viewed as a last resort to save the organelle, a process where damaged mitochondria are fusing with healthy mitochondria to minimize waste and ensure mitochondrial quality control [76,77]. However, once the organelle is beyond repair, mitochondrial autophagy, termed as mitophagy will remove damaged mitochondria. The process of selective degradation of mitochondria by autophagy involves lysosomal degradation of mitochondrial fragments. Key regulators of mitophagy, such as Parkin and PINK1, are involved in preventing diseases such as neurodegenerative disorders [78]. In addition, mitochondrial cargo can also be shuttled from the mitochondrion inside mitochondria-derived vesicles (MDVs) [79,80]. Stunningly, MDVs are of similar size, shape, and cargo content as exosomes. Mostly, ubiquitinated or sumoylated cargo inside the MDVs is shuttled to lysosomes or peroxisomes for degradation [81,82] in a similar manner as the ILV-bearing MVBs. Such similarities point towards similar key role of recycling damaged proteins through generating ILVs and MDVs, as vesicles that encapsulate and deliver unwanted cargo to the lysosome [83,84]. Another shared feature is the lysosomal degradation of MDV and MVB cargo is due to SNARE protein family [85,86].

## 4. Ceramides in EV Biogenesis

### 4.1. Ceramides in EVs Formation

Lipid metabolism and in particular the sphingolipid ceramide is involved in EV formation [6,87]. Ceramide production at the level of the plasma membrane and endosomes regulates EV biogenesis [6,87] (Figure 5). Menck et al. have demonstrated that inhibition of nSMase in human and mouse cells prevents exosome release and significantly increases the release of MVs from the plasma membrane [87]. nSMase and the product ceramide are also involved in the ESCRT-independent pathway for the formation of EVs. Indeed, the internalization of EVs into late endosomes can be triggered by ceramide. Because of their singular cone-shaped structure, ceramides initiate spontaneous membrane invagination that allows for ILV formation into the MVBs and for the maintenance of vesicle shape and structure. In a study conducted by Trajkovic et al., the authors demonstrated that the exosomes were ceramide-enriched and that inhibiting nSMase inhibits exosome release [6]. Moreover, several data indicate that EV miRNA cargo is segregated into distinct subdomains on the endosomal membrane and that transfer of miRNA into the lumen of the endosome required the presence of ceramide [6,88] as well as that proteolipid protein sorting into EVs derived by an oligodendrocyte cell line (myelinating cells of the central nervous system) [6].

The acid sphingomyelinase (aSMase) is also involved in EV formation and release. The acidic enzyme can move onto the cell surface via direct translocation or lysosomal exocytosis in response to stimuli such as CD95 or Pseudomonas aeruginosa infection as well as in wounded cells [89,90,91]. Bianco and colleagues demonstrated that aSMase activity triggers MVs release from glial cells, a process which requires p38 MAPK activation [92]. SMase has been shown to be associated with membrane lipid raft microdomains and its activity results in localized ceramide build-up, thereby creating ceramide-enriched microdomains and inducing membrane curvature [20,28]. In trophoblast cells, alteration of membrane curvature facilitates the budding from the inner plasma membrane to form early endosomes that, when cholesterol-rich, promotes endosome transport for secretion via the MVB pathway for the secretion of EVs [91]. Interestingly, other enzymes of the sphingolipid metabolism such as SMS1 and SMS2 could finely regulate EV formation. The inhibition or silencing of these two enzymes, respectively, lead to the accumulation of ceramide in the trans-Golgi and plasma membrane increasing the release of EVs by microglia cells [93].

### 4.2. Ceramides in EV Secretion

Ceramide has been shown to promote EV production and release in vitro and in vivo. Treatment of several cell lines with exogenous ceramide leads to an increase in the release of EVs [91,94,95]. However, the role of ceramide in EV release is cell dependent and it has been reproduced in several but not all cell lines [2]. Secretion of exosomal miRNAs in HEK293 cells depends on a ceramide-triggered secretory mechanism; on the other hand, inhibition of de novo ceramide formation and nSMase2 did not inhibit the secretion of EVs in PC-3 cells [53,96]. In a recent study, Matsui et al. demonstrated that different EVs are secreted from the apical and basolateral sides of epithelial cells by two independent mechanisms. The apical release of EVs is dependent on the ALIX–Syntenin1–Syndecan1 machinery and the basolateral release of EVs is facilitated by sphingomyelinase-dependent ceramide production [97]. Using mouse model of Alzheimer’s disease (AD), Yuyama et al. showed that treatment and oral administration of plant ceramides promotes the release of neuron-derived EVs with the ability to clear Amyloid-β (Aβ) in neuronal cultures and in mice [98]. Ceramide-dependent upregulation of EVs release may effectively release neuronal EVs and prevent AD pathology [99], suggesting ceramides as a potential therapeutic candidate for AD via secretion of Aβ-enriched neuron-derived EVs.

### 4.3. Ceramide-Enriched EVs

Lipids are important constituents of EVs. Interestingly, the lipid composition of EVs is quite peculiar and is not the same as the donor cells, as EVs are enriched in phosphatidylserine, sphingolipids and cholesterol. A high content of sphingolipids and cholesterol are characteristics of lipid rafts microdomains. Indeed, EVs share these common characteristics of membrane microdomains, particularly lipid bilayer membrane for protection and stability of encapsulated material against detergents [100].

Several pieces of evidence demonstrated that EVs can directly transport lipids from donor cells to recipient cells, altering various cellular processes such as metabolism, release of secretory proteins, and activation of immune cells [101,102]. As previously mentioned, ceramide has a pivotal role in regulating the fate and the activities of cells. For instance, astrocytes treated with Aβ results in the release of EVs enriched with ceramide and pro-apoptosis prostate apoptosis response-4 (PAR-4) [103]. These ceramide-enriched EVs induce apoptosis in recipient astrocytes, which may contribute to the development of AD [103]. Recent data has also indicated increased in serum EVs carrying very long-chain C24:1 C24:1 ceramide in aging and can induce senescence of bone-derived mesenchymal stem cells [104].

The organization of ceramides in the membrane of ceramide-enriched EVs is not fully understood. Elsherbini and colleagues described a distinctive population of EVs that is highly enriched with the sphingolipid ceramide [105]. They proposed that ceramide in the EV membrane was organized in ceramide-rich platforms (CRPs), which are circulating lipid raft microdomain that mediate interaction of ceramide with ceramide-associated proteins (CAPs) [105]. Several stimuli can trigger the release of ceramide-enriched EVs. Treatment with the cytokines (TNF-α and IFN-γ) in an oligodendroglioma cell line induced the formation and release of ceramide (C16-, C24-, and C24:1-Cer species and dihydroCer species) enriched EVs [106]. Treatment of embryonic hippocampal cells with vitamin D3 augments ceramide content in EVs [107]. To date, we do not have an accurate molecular description of the formation of ceramide-enriched EVs.

The levels of ceramide in EVs are associated with various disease conditions [108]. Ermini and co-workers demonstrated that in early-onset preeclampsia syncytiotrophoblast release ceramide-enriched EVs active L-SMPD1 into the maternal circulation [91]. The SMPD1 and ceramide in the EVs promote activation of endothelial cells and impair endothelial tubule formation [91]. Human hepatocytes have also been shown to release ceramide-enriched pro-inflammatory EVs that are involved in macrophage-mediated hepatic inflammation, one of the characteristics of nonalcoholic steatohepatitis [109]. Elsherbini and colleagues demonstrated that serum from a transgenic mouse model of familial AD and serum from AD patients contain ceramide-enriched EVs that are associated with Aβ [105].

## 5. Lysosomes and EVs

Currently there are two main mechanisms involved in lysosomal degradation of ILVs—through its biogenesis inside the MVB or via the autophagy pathway [110,111]. After several cycles of membrane fusion events, an endocytic vesicle will fuse with the early endosome that will later mature into MVB. Once the unwanted cargo starts entering the host cell through endocytosis via the plasma membrane (PM), an early endosome is formed. This cargo will accumulate by entering the early endosome through inwards budding and formation of ILVs. By going through several rounds of ILV formation, the early endosome will grow in size and becomes a late endosome, also known as MVB [112,113]. At this point, the mature MVB is able to fuse with a lysosome. In the majority of cases, this fusion develops an endolysosome whereby the lysosomal lumenal hydrolases initiate a chain of catabolic reactions in order to recycle the protein content of the MVB. In some cases, ILVs inside the MVB are trafficked to the PM and released as exosomes, whereas the majority of MVBs will fuse with lysosomes to form an endolysosome [2].

### 5.1. Lysosomal Endocytosis (LE) and EV Release

Once the cargo is packaged into the ILV, it is considered for elimination due to its unwanted cargo. Although lysosomes are considered as terminal organelles that systematically degrade and recycle cargo, some of the cargo can escape through fusion and release of the lysosomal content through the PM [114,115]. Once the MVB fuses with the lysosome, the processing of the MVB cargo does not appear start immediately, as there are reports that ILVs can be released during a process called lysosomal exocytosis (LE). LE is involved in maintaining cell homeostasis. The release of LE derived vesicles is necessary to prevent disorders, such as lysosomal storage disorders, fibrosis, neurodegenerative diseases and neoplastic transformation of the cell. In these LE-derived vesicles, intact ILVs are exosomes that are loaded with potentially harmful ubiquitinated cargo [5,110,116]. Organelles, such as MVB and autophagosome, can shift from fusion with the lysosome to fusion with the PM for extracellular release of its content. Interestingly, autophagosomes can also fuse with the late endosomes (also known as MVB) and produce amphisomes which later fuse with the lysosome or fuse with the PM for release of the contents. The MVB-autophagosome-lysosome crosstalk is highly dynamic, as it can shift the cargo from degradation to exocytosis release, and vice versa [110]. One such protein involved in the MVB-autophagosome-lysosome axis is S1P, a pro-survival and anti-apoptotic factor that opposes ceramide pro-apoptotic activity [117,118,119]. The S1P receptor, called G protein (Gi)-coupled sphingosine 1-phosphate (S1P) receptor, is involved in MVB maturation as well as cargo sorting into the ILVs [119]. Several autophagy genes, such as Atg5 and the Atg12–13 complex, regulate the release of exosomes. Such decrease or increase in exosome release can be due to drug-stimulated pH increase and further LE [120]. Furthermore, the interaction of Alix, an ESCRT component, with Atg12-Atg3 is critical during early stages of autophagy [121,122].

LE is a ubiquitous Ca^2+^-dependent mechanism involved in PM repair [123], neuron protection and prevention of neurodegenerative diseases [110,124], immune response and antigen presentation [125,126] and intracellular Ca^2+^ induced release of exosomes [127,128]. A unique role of LE is to regulate ATP levels due to its potential toxicity in excess amounts [129,130]. More interestingly, LE is closely linked with PM repair in order to restore the natural barrier and selective permeability of the cell. Lysosomes located near the PM damage site will rush to relocate to restore PM integrity [131]. Once the cell is ruptured, a massive Ca^2+^ influx activates a synaptotagmin VII (SynVII)-dependent mechanism of PM repair. During this process, lysosomal proteins VAMP7, C2B and C2A will assist SynVII in disassembling and restructuring the lysosome in order to integrate it into the damaged area of the PM. The restructuring process also involves the interaction with the PM surface-protein SNAP-23 and Syntaxin 4 [132,133]. Once a complex of Syntaxin 4/VAMP7/SNAP-23 is formed (also known as the SNARE-complex), the lysosome will dock with the PM [134]. Such docking requires increased Ca^2+^ levels, which subsequently will trigger interaction between SynVII and SNARE-complex [135,136]. A patch protein complex, in the presence of Ca^2+^ and the lysosomal calcium channel mycolipin-1 (TRPML-1), will ensure a smooth transition of the lysosomal content with the PM [111,137,138]. Furthermore, acid ceramides are critical for TRPML-1 channel-mediated Ca^2+^ release, which controls lysosome-MVB fusion and exosome release in podocytes [111]. Lysosomal Ca^2+^ release through TRPML-1 activates calcineurin, which will bind and de-phosphorylate the transcription factor EB (TFEB). De-phosphorylation of TFEB will promote its nuclear translocation, which is linked with LE and autophagy, as well as proper lysosomal cargo clearance and fusion between the lysosome and PM [139,140]. A delay in ILV degradation inside the lysosome can later result in exosome release during LE [141], which directly links exosome release with TRPML-1-mediated LE [142]. Furthermore, a delay in tumor ILV lysosomal degradation can induce cell migration via activating the MAPK pathway [143]. All this highlight the role and importance of lysosome and LE in exosome release, uptake and degradation.

The processes regulating lysosome function and exosome release are tightly linked. Lysosomal alkalized pH is linked with impaired lysosomal function and an increase in exosome release [144,145]. An increased release of exosomes is associated with several lysosomal storage diseases (LSDs). There is evidence that lysosomal dysfunction by stored material can be the cause of ILV release from MVB via PM-MVB fusion in order to ensure cellular homeostasis [146]. A major viral oncoprotein latent membrane protein 1 (LMP1) can activate mammalian target of rapamycin complex 1 (mTORC1) in order to suppress host autophagy and favor cellular growth and proliferation. Interestingly, LMP1 and mTORC1 activation also opposes exosome release [147,148,149]. Inhibition of the mTORC1 component mTOR1 induces Ca^2+^ release through TRPML-1. This Ca^2+^ release is regulated by transmembrane BAX inhibitor motif containing 6 (TMBIM6), a Ca^2+^ channel-like protein. This increase in lysosomal Ca^2+^ activates PP3/calcineurin and trigger TFEB translocation into the nucleus, leading to autophagy and lysosomal biogenesis [150]. The classical exosome biomarker CD63, one of the key factors in exosome production and endosomal cargo sorting, coordinates endosomal packaging and autophagy. A reduction in CD63 is associated with development of autophagosomes. This means that the balance shifts from ILV-lysosome cargo degradation towards MVB-PM fusion and CD63+ exosome release [147,151]. However, it is still unclear if the exosomes escaped before or after the MVB is fused with the lysosome.

LE is important for PM remodeling and reorganization. LE is particularly vital in phagocytic cells, such as macrophages when they rapidly reassemble actin under stress situations. Macrophages can internalize large amounts of lipids [152,153,154]. Actin reorganization is driving the whole process and contributes in filopodia, lamellipodia and other processes of membrane ruffling. Such phagocytosis necessitates considerable intracellular membrane relocation and LE is the central process for reorganization and integration of the membrane [154,155]. Another study found that LE is associated with increased exosome release [141]. LE is negatively regulated by neuraminidase 1 (NEU1), a sialidase mutated in the glycoprotein storage disease sialidosis. The complete deletion of NFU1 triggers LE and increased exosome release. These exosomes carry ubiquitous cargo, such as misfolded transforming growth factors, and could cause pathogenesis. These exosomes, once taken up by normal fibroblasts, convert them into myofibroblasts with altered proliferative and migratory properties [141]. In addition, phagocytic macrophages are sensitive to lysosomal dysfunction and consequently use exosome release to get rid of unwanted cargo, which can downstream regulate immune responses or contribute to disease progression [156,157]. In summary, there is clear evidence that the lysosome has a central role in exosome release under stress and LE conditions. Furthermore, such LE-regulated release of exosomes can contribute to initiation or progression of certain diseases.

### 5.2. Lysosome and EV Uptake

EVs are found inside the lysosome of recipient cells as soon as 1 h after uptake [158,159]. The mode of uptake in phagocytic cells, especially macrophages, is likely through pinocytosis and phagocytosis [160,161,162,163]. Smaller and larger particles (~100 nM and >500 nm, respectively) are internalized through phagocytosis [164]. As an example, leukemia cell-derived exosomes can be taken via DNM2-dependent phagocytosis and co-localize with phagolysosome biomarkers, which indicates that foreign exosomes are likely to be recycled and eliminated [165,166]. During pinocytosis, smaller particles (<100 nm; micropinocytosis) or larger particles (up to 5000 nm; micropinocytosis) are taken up [167,168]. Pinocytosis of these vesicles can be facilitated through the cooperation of caveolin and/or clathrin [169,170], proteins that are vital for endosomal budding and packaging. The blocking of clathrin-mediated endocytosis inhibits EV uptake [161,171]. Similarly, knockdown of calveolin impaired the uptake of EVs [159]. Macropinocytosis, on the other hand, allows for a larger number of EVs to be taken up [162]. Macropinocytosis is a non-selective uptake, and like any pinocytosis will likely involve degradation of the EV cargo once it’s inside the cell through the proteasome or the lysosome [172,173,174].

### 5.3. Lysosomal Storage Diseases (LSDs)

LSDs are a group of about 70 rare genetic conditions that cause a toxic buildup of substrates that damages cells and organs due to defective functioning of the lysosome. LSDs are more common during pregnancy or soon after birth, with adult LSD cases less severe than newborns. To date, there is no cure for the various LSDs, with current treatment lessening the damage. LSD state occurs once undigested protein cargo is accumulated inside the lysosome-endosome system due genetic mutations or pharmaceutical treatment that led to deficiency of a specific enzyme required for digestion of glycoproteins or lipids [175]. Such lysosomal dysfunction is the cause of the rise of several rare diseases, depending on the route and type of altered protein cargo that is accumulated inside the cell. The most frequent disorders are associated with the mitochondria-lysosome axis. Intracellularly, lysosomal Ca^2+^ is released by TRPML1, and enters transfers to mitochondria once the mitochondria-lysosome contact is established. Loss of TRPML1 alters the mitochondria-lysosome axis and results in the LSD mucolipidosis type IV [176]. TRPML1 activation is critical for TFEB-mediated LE and lysosomal cargo clearance, which makes TRPML1 a promising target for treatment of various LSDs [139]. Other disorders, such as neurodegenerative Parkinson’s disease, ones, occur due to improper autophagic cascade and excessive overwhelming of lysosomal potential [177,178]. Strikingly, LSDs are linked with unusual and altered exosome release and uptake, and TRMPL1 and other key players of the LE represent promising targets for therapeutic strategies [142]. Owing to its importance in endosomal transport, the targeting of the cytoskeletal system that carry exosomes to the perinuclear region for further fusion with the lysosome is another promising strategy [171].

### 5.4. Lysosomal Dysfunction and EVs

Lysosomes are acidic organelles that degrade and recycle cargo mainly via autophagy or endocytosis, respectively. They also play important role on regulation of autophagy, mitophagy, and other protein-degradation processes [79,179]. Several diseases can arise due to lysosomal dysfunction, as exemplified by the neurodegenerative disorder [180]. In other diseases, such as AD and Parkinson’s, toxic proteins are aggregated inside EVs that once released, contribute to inflammation and disease progression [181,182,183]. Presence of key proteins, such as DJ-1, Parkin or PINK1 are all involved in lysosomal dysfunction, mitophagy, and neurodegenerative diseases [184,185]. Parkin, a protein involved in Parkinson’s disease, is vital for modulating endosomal organization and function of the endosomal-lysosomal pathway [186]. Mitophagy is highly dependent on PINK1 and Parkin, two key players in Parkinson’s disease, which involvement is regulated by exosomes. The exosome-mediated PINK1/Parkin-dependent mitophagy is important for intercellular communication in impaired neurodegeneration-involved cells [184]. Exosomes can carry several biomarkers associated with diseases. For example, several exosomal biomarkers are associated with AD, particularly neurotoxic amyloid-beta (Aβ), lactoferrin, Tau, Acetylcholine esterase (AChE), and amyloid precursor protein (APP). These exosomal proteins, particularly increased levels Aβ and Tau, further propagate disease progression of AD [187,188]. Similarly, in Parkinson’s disease, DJ-1, alpha-synuclein (a-syn), AChE and Tau, represent key exosomal biomarkers. Mostly detected in the plasma, DJ-1 and a-syn are prominent in exosomes of PD patients [189]. Such packaging of a-syn can be promoted by Secretory Carrier Membrane Protein 5 (SCAMP5), a protein regulated by TFEB. SCAMP5 inhibits autophagy flux by blocking the fusion of autophagosomes with lysosomes [190,191]. This exosome release has been shown to be Ca^2+^ -dependent [192]. Interestingly, a similar exosome mechanism can be observed in Huntington’s disease, which involves mutated huntigtonin (mHtt)-regulated decrease secretion of exosomes. The progression of Huntington’s disease may be inhibited through the administration of astrocyte-derived exosomes, which reduced the density of mHtt aggregates. In general, the study found that mHtt reduces the expression of αB-crystallin in astrocytes and decreases exosome secretion [193]. During Amyotrophic Lateral Sclerosis (ALS), neurons release exosomes and initiate motor neuron death. ALS-exosomes carried TDP43 and TSG-101, whereas cells had mutated TDP-43, as well as several mutations of the autophagy pathway. Exosomes carrying TDP43 can be the potential cause of TDP43 aggregation and TDP-43 proteinopathy and are linked to autophagy inhibition [194]. In all cases, autophagy, mitophagy and lysosomal activity are key factors that prevent the MVB rupture and distribution of exosome for ALS initiation and progression [188,195]. Exosomes containing these proteins can potentially be used as early disease biomarkers, such as for detection of cancer [196,197]. Furthermore, lysosomal exocytosis during cellular neurodegeration will enable release and potential detection of exosomes carrying biomarkers [124].

### 5.5. Cancer, Exosomes and Lysosomal Dysfunction

Exosomes carry specific cargo that can contribute to carcinogenesis. Several types of molecules, such as miRNA, are associated with tumor arise and progression, and are packaged into exosomes [1]. Since lysosomes are involved in numerous cellular processes, such as programmed cell death, antigen presentation, cell adhesion/migration and exosome release, there is a tight connection of this organelle with neoplastic transformation of cells. Lysosomes were shown to help cancer cells to survive stress due to lack of nutrients via autophagy [198]. Via adjusting the intracellular pH, lysosomes can also regulate degradation of CTLA-4, a key player in cancer cell evasion of T-cell specific response [199]. Acidic environment will affect cancer cell activity. However, via lysosomal exocytosis cathepsins are released and will promote cancer metastasis [200]. In cancer cells pH shifts will affect the ESCRT system. The MVB will move towards the positive end of the microtubule, and avoid lysosomal degradation in favor of fusing with the PM for exosome release, whereas alkaline pH reduced exosome secretion [201]. These exosome will carry tumor-associated miRNA with metastatic potential [202]. Under tumor conditions, the lack of nutrients will result in starvation for the rest of the normal cells linked with autophagy and endosomal pathway disorders [122]. Once extracellular content, including exosomes, are endocytosed, phagocytosed or pinocytosed, the content is delivered to the lysosome for breakdown and nutrient generation. In cancer cells, the nutrient microenvironment is depleted, and will require sacrifice in the form of healthy cell components [203]. All this links lysosomal dysfunction and exosome release with neoplastic transformation and metastasis.

## 6. Conclusions Remarks

EVs are a unique class of cell-to-cell communication whereby cells secrete a variety of macromolecules to distant cells. There is a growing appreciation of the precise role of EVs for cellular homeostasis and defects in EV biogenesis, cargo sorting, release, and uptake is the underlying cause of many disorders. In this review, we have highlighted the role of ceramides and lysosomes in the biology underpinning EV-based communication. While we are gaining deep insights of our knowledge of EVs, much remains unknown about the regulatory role of lysosomes and ceramides in physiological and pathological conditions, including the underappreciated heterogenous and diverse nature of vesicles. Nevertheless, we are at the point where we can exploit these pathways for therapeutic use of various vesicle associated disorders.

## Figures and Tables

**Figure 1 ijms-23-15317-f001:**
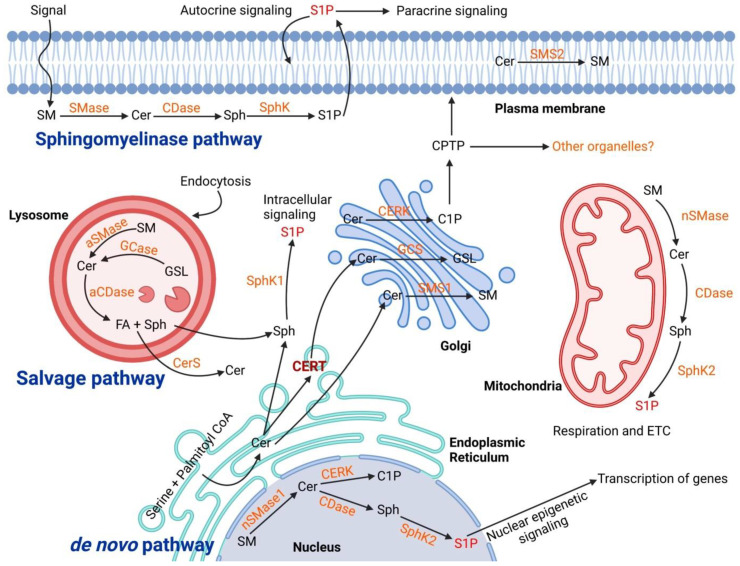
Biogenesis of ceramide in cellular organelles and the complex biochemical pathways in downstream. The ceramide main synthesis routes are De novo in the Endoplasmic Reticulum, Salvage pathway in Lysosome, and Sphingomyelinase both in lysosome and plasma membrane. The transfer of ceramides to the Golgi is mediated through ceramide transfer protein (CERT) within vesicles. In the Golgi, different classes of sphingolipid are synthesized and transported to the cellular sites of action. In the plasma membrane and lysosomes, the main catabolic pathways of sphingolipids are “sphingomyelinase (SMase)” to produce ceramides. The produced ceramides go under further metabolism to generate other sphingolipid species such as sphingosine (Sph), sphingosine-1 phosphate (S1P) and also sphingomyelin (SM). Abbreviation: Acid Ceramidase (aCDase), Ceramidase (CDase), Ceramide Kinase (CERK), Ceramide Synthase (CerS), Glucocerebrosidase (GCase), Glucosylceramide Synthase (GCS), Glycosphingolipid (GSL), Sphingomyelin Synthase (SMS).

**Figure 2 ijms-23-15317-f002:**
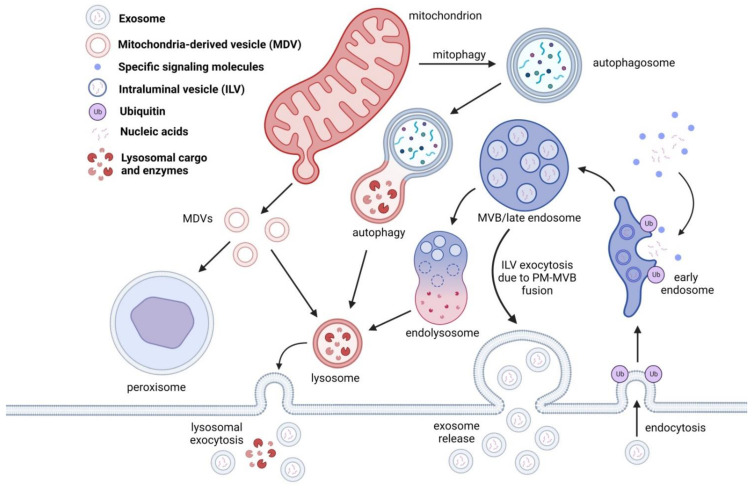
EVs are generated through the endosomal pathway. Once the early endosome is formed, ILVs are packaged with proteins destined for degradation, nucleic acids, and other signaling molecules. Once the cargo is tagged for degradation, an endolysosome is formed and will proceed to lysosomal degradation. In some cases, the ILVs are released during MVB-PM fusion and are consequently released as exosomes. In some cases, lysosomal cargo containing ILVs can also release exosomes through lysosomal exocytosis. Lysosomes are also involved in protein degradation through autophagy, mitophagy and mitochondria derived vesicles (MDVs). MDVs, unlike MVBs, are targeted for peroxisomal cargo degradation.

**Figure 3 ijms-23-15317-f003:**
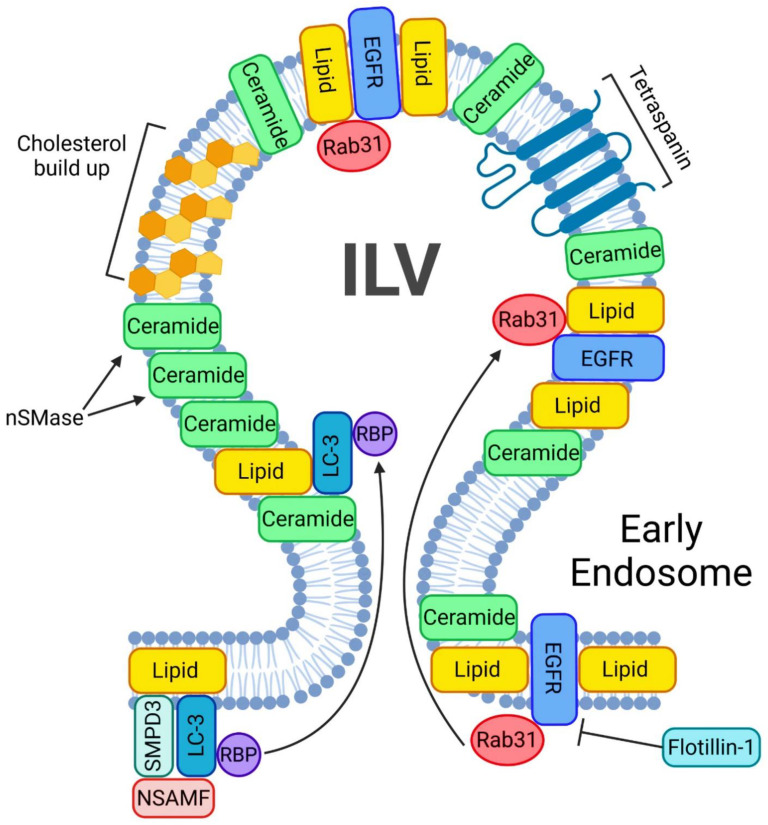
ESCRT-independent pathway of exosome biogenesis and release. A crucial step in starting the ESCRT-independent exosome biogenesis is the phosphorylation of Rab31. A phosphorylated Rab31 will allow flotillins to deliver the EGFRs into the ILV. nSMases will help accumulate ceramides inside the ILVs. In addition, such ILVs will be enriched in ceramides and cholesterol. Another mechanism of lipid buildup inside the ILV involves LC3, which also incorporates RNA binding sites and small non-coding RNAs.

**Figure 4 ijms-23-15317-f004:**
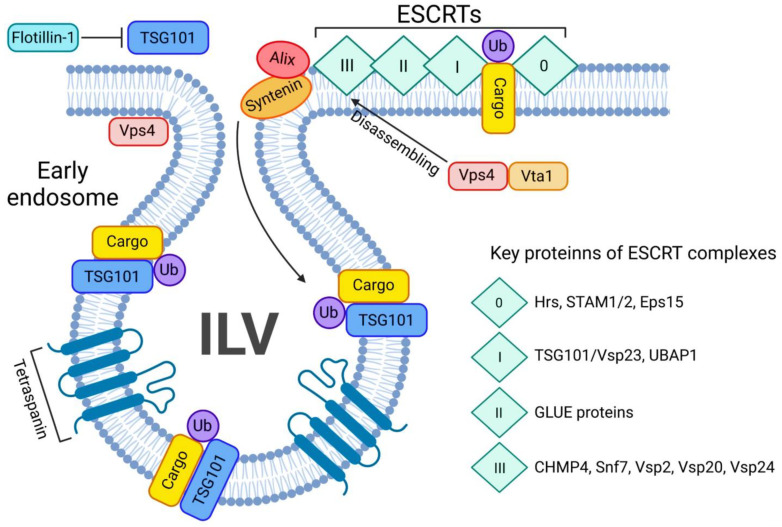
ESCRT-dependent pathway of exosome biogenesis and release. The ESCRT-dependent exosome biogenesis starts with assembling all 4 ESCRT complexes on the ubiquitin-bound cargo (Ub-cargo). The process is tightly regulated by key proteins such as Alix and syntenin. Once ESCRTs bind the Ub-cargo, TSG101 navigates the cargo inside the ILV and the Vsp4/Vta1 complex will disassemble the ESCRT 0-III complex. Tetraspanins will favour the budding of the ILV.

**Figure 5 ijms-23-15317-f005:**
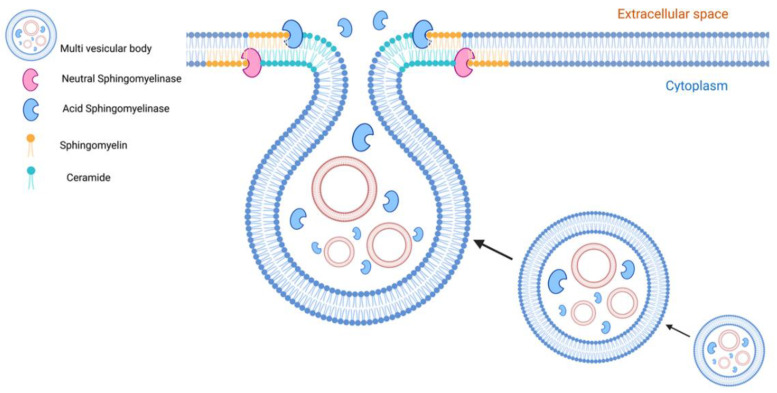
Role of sphingomyelinases and ceramides in EV formation. Sphingomyelinases are active at the level of endosomes and cell membranes whereas their natural products, ceramides, are involved in the biogenesis and secretion of EVs. Neutral sphingomyelinase (nSMase) and ceramides participate in the endosomal formation of exosomes in an ESCRT-independent pathway. Acid sphingomyelinases (aSMase) targeted directly or indirectly via lysosomal exocytosis to the cell membrane contribute to creating ceramide-enriched microdomains that are involved in microvesicle and exosome formation and secretion.

## Data Availability

Not applicable.

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
