# Peer review of "Role of Ceramides and Lysosomes in Extracellular Vesicle Biogenesis, Cargo Sorting and Release"

_ijms, 2022, doi:10.3390/ijms232315317_

Round 1
Reviewer 1 Report
The review by Horbay and colleagues titled "Role of Ceramides and Lysosome in the Extracellular Vesicles Biogenesis, Cargo Sorting and Release" is full of useful and interesting content. The topic is of great scientific interest, given the fundamental role attributed to extracellular vesicles in numerous physiological and pathological mechanisms. The study very accurately clarifies the importance of the role of ceramide and lysomes in functions related to extracellular vesicles.
However, I have some suggestions that I think could make improvements to this interesting manuscript.
1) The introduction is a bit short and looks like an abstract, however it summarizes all the contents of the text, but there are no bibliographic references, please add them to the text.
2) Ceramides:
- I suggest to open paragraph 2.3 "Biogenesis of Ceramides: Role of Endoplasmic Reticulum, Golgi and Lysosome" before paragraph 2.2 "Physiological Function of Ceramides"
- I suggest changing the title of paragraph 2.2 to "Physiopathological Function of Ceramides"
- Line 127: add the acronym of Fatty Acid (FA).
3) Extracellular vesicles:
- Despite the clear explanation given in lines 261-263 and although similar images are already reported in other works, for greater clarity I suggest to insert in addition to figure 2 also two separate images, one on EVs biogenesis ESCRT-dependent and one on EVs biogenesis ESCRT- independent.
- Line 238: replace the DOI with the bibliography number
4) Ceramides in EVs Biogenesis:
- Line 295: adding a space between exosome and release.
5) Lysosome and EVs:
- Line 417: replace (Tardif et al. 2013; Trombone et al. 2007) with bibliography numbers.
- Lines 565-567: add bibliographic references.
Reviewer 2 Report
The review shed of light to role of ceramides and lysosome in the EVs biogenesis, cargo sorting and release from cell. Despite the high relevance, this topic is poorly covered in the literature. The article is written in good language. My comments are minor:
-remove minor typos (Evs, merged words, etc)
- move the drawings to where the authors mention them
- correct the list of references according to the rules of the journal
I think that manuscript entitled "Role of Ceramides and Lysosome in the Extracellular Vesicles
Biogenesis, Cargo Sorting and Release" should be accepted for publication in International Journal of Molecular Sciences after minor revision.
